# Case Study of Remodelling the As-Built Documentation of a Railway Construction into the BIM and GIS Environment

**Dalibor Bartonek** [1,*] **, Jiri Bures** [1] **, Ondrej Vystavel** [1,*] **and Radomir Havlicek** [2]

1 Institute of Geodesy, Faculty of Civil Engineering, Brno University of Technology, Veveri 331/95, 602 00 Brno, Czech Republic
2 Railway Administration, State Organization, Dlazdena 1003/7, 110 00 Prague, Czech Republic
* Correspondence: bartonek.d@fce.vutbr.cz (D.B.); vystavel.o@fce.vutbr.cz (O.V.)

**Featured Application: The solution was applied for the remodelling of the as-built documentation of railway constructions of Railway Administration of the Czech Republic.**

**Abstract:** Building Information Modelling (BIM) is a modern approach to managing the process of preparation, realization and operation of building objects including their documentation throughout their life cycle, based on database agenda platform. The aim of our research is to analyze and innovate existing engineering procedures with the aim: 1. to remodel the existing CAD documentation into BIM for the purpose of public procurement, 2. to provide guaranteed data to the IS of the Digital Map of Public Administration and 3. to provide data for the design of new railway structures or their reconstruction. The aim of the case study was to evaluate the effectiveness of remodelling the existing as-built documentation of a railway construction into a common BIM data environment (CDE), in which further subsequent construction agenda should be managed for the remaining period of its life cycle. Using the documentation for construction realization of the railway station Šumice, this 3D documentation was remodeled into the BIM data environment CDE and alternatively also into the 2D GIS environment. The BIM data standard developed by the State Fund for Transport Infrastructure was analyzed during the documentation reworking. An important parameter of the documentation rework was the use of a geodetic reference system fully compatible with the cadastral system in the Czech Republic. It turned out that the general data standard is only partially applicable for railway structures containing many special objects and many objects requiring individual classification. The remodelling of existing graphical data proved faster and more efficient in a GIS environment (layer oriented) compared to the need for 3D remodelling in a BIM CDE (object oriented). Experimental results have demonstrated the effectiveness of remodelling underground technical infrastructure objects, while the visible surface situation is often more effectively captured by current progressive bulk data acquisition technologies. In a CDE environment, existing as-built documentation data can be efficiently stored and administered and progressively, for the procurement and execution of construction, purposefully remodeled only to the extent required in BIM or converted into an exchangeable Digital Technical Map (DTM) format for public administration.

**Keywords:** BIM; CDE; GIS; remodelling; as-built documentation; railway station

## 1. Introduction

If we want to live in smart cities and get around safely, we should be able to build smart, and building smart means building in Building Information Modelling (BIM). Information in BIM must be digital, accurate, shared, updated, coordinated, pictorial, geometric and descriptive [1]. By stakeholders sharing such information, BIM enables effective Building Information Management of buildings. This problem is solved, for example, in [2]. A model is digital representation of a building in structured form containing geometric, technical and other non-geometric data. The digital model (replica, digital twin) allows to

detect possible errors and collisions already in the design phase and thus to prevent errors during the realization of the construction. Thus, the building exists in the phases designed, under construction, in operation, reconstructed and to be demolished. It means that it is necessary to ensure transitions from digital form to reality and vice versa. BIM is therefore a collaborative method that, by sharing information on a common repository and using digital modelling, enables all disciplines to work together effectively, helping to ensure the optimization of the construction process and the efficient operation of the building.

In 2017, the Government of the Czech Republic, by its Resolution No. 682, approved the Concept for the introduction of the BIM method in the Czech Republic [1]. The Ministry of Industry and Trade (MIT) has been entrusted with the responsibility for the implementation of BIM in the Czech Republic. Within the BIM system, digital multidimensional models of buildings containing geometric and descriptive information are to be created, which will serve as an open database of information about the building for its design, execution, operation and interconnection of these stages. The use of BIM is an essential condition for the digitization of the construction industry in the Czech Republic. Within the department of transport, the MIT is cooperating with the State Fund for Transport Infrastructure (SFTI) on the implementation of BIM to process of construction. From July 2023, public procurement authorities will be obliged to use BIM for all public procurement contracts for construction works budgeted above EUR 6.25 million, including the elaboration of their preparatory and design documentation, considering the specifics of individual types of construction.

The Railway Administration (RA) is a state investor organization in the field of railway constructions, also playing the role of the railway infrastructure manager. In the Czech Republic, EUR 5.5 billion was invested in transport infrastructure in 2022, with approximately 50% of the total sum invested in railway infrastructure. The Czech Railway Administration manages approximately 9600 km of railway lines. In connection with the introduction of BIM, the conversion of existing construction documentation in 3D vector CAD format to the BIM Common Data Environment (CDE) is being addressed.

The existing documentation of railway structures within the Railway Administration of the Czech Republic (RA) is in 3D CAD format and in the area of the so-called railway perimeter, specified by the Railway Act [3], which represents the area on both sides of the railway defined by vertical areas led by land boundaries (approximately 10 m on both sides from the track axis) on the total length of 9600 km of railway lines in the Czech Republic.

The Railway Administration wants to remodel this documentation in connection with the Public Procurement Act, which requires the use of the BIM platform. Also, in connection with the creation of the Digital Technical Map of the Czech Republic (DTM), which is a part of the information system of Digital Map of Public Administration, according to the amendment to the Surveying Act [4], which requires the use of a GIS platform.

At the same time, the Railway Administration wants to develop methods of effective data transfer between the two platforms, each of which has different contents and structure. The Railway Administration also wants to provide guaranteed data for the design of new railway constructions or for their reconstruction to designers and also to the public administration through the transfer of selected data to the DTM.

The aim of the case study was to assess the effectiveness of remodelling the existing as-built documentation of the railway construction into a common BIM data environment (CDE) and alternatively into a GIS environment, in which the further subsequent agenda of the construction should be kept for the remaining period of its life cycle.

Another objective was to identify, through a case study, the existing problems of this remodelling and to propose innovations to the existing engineering procedures and internal regulations that would lead to meeting the above requirements of the Railway Administration of the Czech Republic.

BIM models of buildings must be usable for updating the digital technical map (DTM) containing elements of the planimetry (buildings, traffic roads, technical infrastructure networks) and hypsography. DTM is produced by local authorities and transport and technical infrastructure managers for their own use. DTM is unique in their parameters,

especially in their content and accuracy. The accuracy of the position and height data of the sub-elements of the DTM contents is characterized by the standard coordinate deviation $\sigma_{xy}$ and the standard height deviation $\sigma_H$. The accuracy data is given for the element or for the individual points of the spatial determination in the form of the accuracy class specified in the Czech legislative regulation designated as Decree No. 393/2020 Coll. on DTMs. The accuracy is based on the Czech technical standard ČSN 01 3410. In addition to the key data, DTM also consists of the relevant information and communication technologies, i.e., hardware and software. In addition to those who acquire it, DTM serves citizens and public administrations as well as private companies, in particular owners and operators of technical infrastructure, planners, investors, banks and insurance companies. The DTM contributes to a better awareness of the land, and the fact that there is no potential problem hidden beneath its surface helps with pre-project preparation, allowing for communication with the relevant authorities digitally. For mayors of municipalities or city districts, up-to-date data maintained in the DTM helps to efficiently resolve operational situations, prevent neighbor disputes, as well as ensure safety and protect the lives of citizens. For example, emergency situations on underground technical infrastructure facilities can be eliminated in a much shorter time using the DTM by not wasting time searching for the necessary documents to eliminate the relevant malfunctions.

The DTM allows to work with uniformly processed data on all types of transport and technical infrastructure, to digitalize the processes of construction management, to simplify and streamline the preparation and permitting of buildings, to streamline the planning of buildings and investments and also to minimize the administrative burden, to keep better passports of municipal property, to organize unsettled property relations by comparison with the cadastral map, to streamline a number of public administration activities, to manage the administration of the built-up area in a modern way and to solve a large number of life situations of citizens.

### 1.1. The Construction Process in the Czech Republic and Its Documentation

The process of construction, especially of large investment units (motorways, roads, buildings with spatial composition, industrial buildings, flat buildings, etc.), is complex in the Czech Republic and is accompanied throughout its course by extensive documentation. Placing buildings, changing the use of the territory and protecting important interests in the territory can only be done on the basis of a planning permission issued by the building authority. Construction, alterations and maintenance work can only be carried out under a building permission or on the basis of a notification to the building authority. The entire construction process consists of several interdependent phases. It includes the preparation phase, the realization phase and the operation phase. The legal participants in the preparatory phase of construction are the investor, the competent authorities and the owners of property affected by the construction. The contractual parties in the construction phase are the investor, the designer and the contractor (association of contractors). The documentation in the construction process consists of written and graphic parts. The construction process is the integration of two distinct but mutually reinforcing components (aspects):

1. normative part—legal aspects and documents
2. material part—technical aspects and documents

The normative part of construction activity includes general (Building Act No. 183/2006 Coll. and a total of 10 implementing decrees to it, of which Decree No. 499/2006 Coll. directly regulates the building documentation), and special (relevant technical standards) regulations including necessary construction and other related documentation (construction drawings). The material part includes the set of materials, technological equipment and methods that are strictly necessary for the realization of a specific construction.

The aim of the application of GIS or BIM platform tools in the construction industry is the possibility of uniform integration of data from the entire construction process and thus achieving their effective availability throughout the entire construction process, subsequent

use of the building including the possibility of use for subsequent future reconstruction. The solution to the problem is to link all these key elements of the whole process including:

1. generally applicable documents (legal regulations and technical standards),
2. building documentation,
3. factors affecting construction in the area (geographical and climatic conditions, etc.),
4. realization factors (construction management).

The entire construction process in the Czech Republic can be plotted on a timeline and the basic context can be related. In the preparatory phase, the investor lets the designer prepare his ideas about the construction in a documentation of the type of investment plan or study, using publicly available information about the territory in the form of territorial analytical documents (TAD) at scales from 1:500,000 to 1:1000, hence maps.

The following Figure 1 shows the sub-phases of construction on the timeline in the context of the corresponding documentation type. The usual scales of the graphic documents used and the graphic drawings produced are also shown. From the point of view of the homogeneity of the geodata, an important factor is also their accuracy, which in the preparatory, design and operational phases corresponds to the accuracy for the large-scale state mapping work, i.e., the standard of digital cadastral maps; in the documentation for assignment of the construction (Tender Design—TD), the technical and qualitative conditions define the parameters of the geometric accuracy of the construction and this is subsequently implemented in the realization phase. TAD accuracy means the accuracy $\sigma_{xy} = 0.14$ m, Geometric accuracy means the accuracy of geometric parameters stated in the relevant technical standards of geometric accuracy and usually ranging from several mm to cm. In the implementation or operational phase, sub-millimeter accuracy is required in connection with the measurement of displacements and deformations.

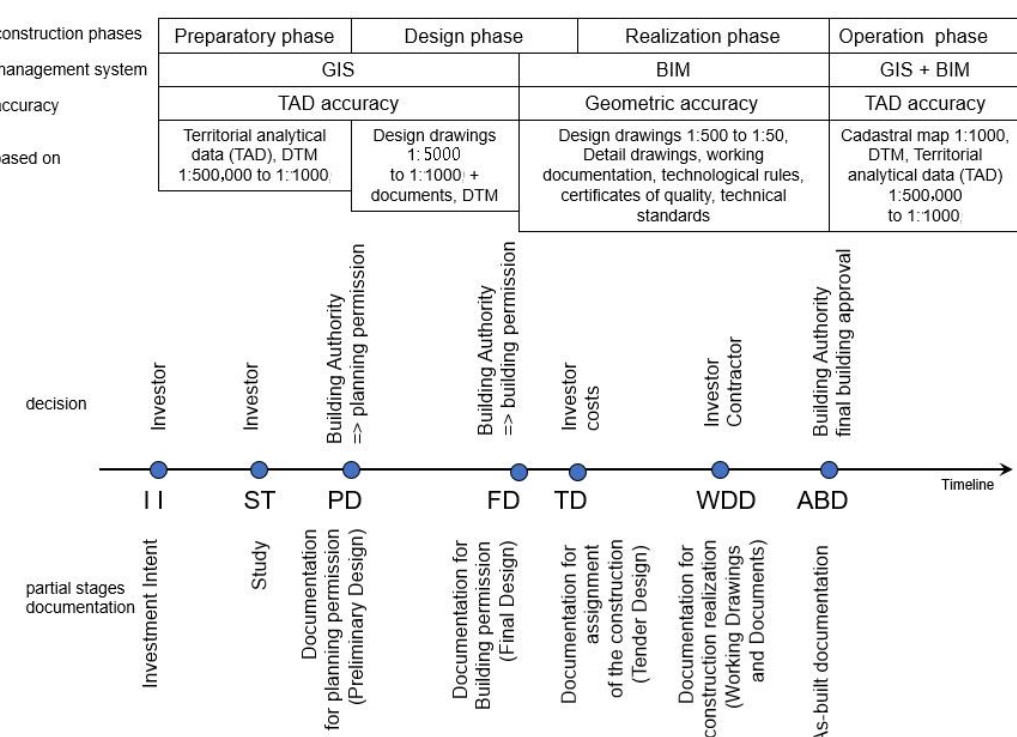

**Figure 1.** Sub-phases of the construction process and their documentation on the timeline.

If the data is continuously stored in the unified data warehouse of the Building Information System, then it is very effectively structured and accessible within the defined roles of the construction participants. A specific feature of the Construction Information System is the internal data structure divided into spatially bound documents, legal regulatory documents—generally valid, local, time-dependent.

During the construction process, different types (levels) of documentation are prepared according to the purpose (Figure 1):

The main purpose of the documentation type "Investment Intent" (II) is the initial materialization of the investor's conceptual ideas about the building, especially in terms of purpose, spatial layout, location of the building in the area and its connection to the technical infrastructure.

The documentation type "Study" (ST) is a subsequent elaboration of the investment plan into alternative solutions, whereby the optimal design and alternative designs are evaluated.

This is followed by the preparation of documentation for planning permission, which is submitted to the building authority, which then initiates a public building approval process with the affected parties.

The "documentation for the planning permission" (PD) considers the location of the building within the limits of the territory, and within the planning procedure it is necessary to enter into negotiations with the relevant authorities, technical infrastructure administrators and other participants. An important part of the documentation is the documentary part, which contains statements from the relevant authorities, technical infrastructure managers, expert opinions and other statements. The planning decision-making process ends with the issuance of a planning permission, which specifies the restrictive conditions for the location of the construction in the area that resulted from the planning procedure in the context of the investment project [2].

Next comes the design phase, in which the investor commissions the designer to prepare the design documentation for the building permission.

"Documentation for building permission" (FD) is a technical elaboration of the PD in terms of the restrictive conditions specified in the planning permission, the subject of the solution is the specific spatial location of the building and layout, connection to public technical infrastructure and roads, the layout of the construction site and the principles of construction organization. The FD includes the drawing and documentary part of the documentation prescribed by the Building Act, which is submitted to the building authority. The building authority initiates a public building procedure process, which includes consultation of the construction with the affected parties. The building permission process ends with the building authority issuing a building permission, which specifies the purpose and description of the construction and sets out the binding conditions for the execution of the construction in the context of the constraints resulting from the building procedure process.

The design phase ends with the preparation of the "Documentation for assignment of the construction" (TD), in which the TD is supplemented by the investor's Technical Quality Conditions (TQC) and other conditions of the public tender. The TD serves as the basis for the public procurement procedure for the construction contractor, for the determination of the construction price and for the evaluation of the bids of the bidders.

The winner of the public tender becomes the construction contractor, who implements the construction by technical means on the basis of the prepared "Documentation for Construction Realization" (WDD), which is a detailed technological documentation, e.g., production and assembly documentation, technological procedures, etc. The WDD is prepared by the contractor or the designer on the order of the contractor.

The realization phase ends after the completion of the construction with the so-called approval of the building, which is issued by the building authority. The progress of construction is continuously documented in the "As-built Documentation" (ABD), the aim of which is to document any changes to the WDD and FD, to demonstrate the implementation of the construction within the prescribed parameters and at the location prescribed by the project in accordance with the zoning decision and building permit according to the Building Act No. 183/2006 Coll. The ABD includes construction drawings corrected according to the as-built state of the construction objects and the geodetic documentation including position and elevation survey of the construction objects according to reality

with assessment of their actual spatial position. In these works, optimization methods are often used in connection with GIS [5]. The ABD together with other documents, certificates is submitted for approval to the building authority, which assesses the compliance of the implemented construction with the conditions of the planning permission and building permission. The building process ends with the issuance of the approval and the building can be used. From the documentary point of view, the ABD concludes the construction process and serves to update the digital technical map (DTM) and subsequently the data in the TAD.

*1.2. Implementation of BIM Standards in the Czech Republic*

The introduction of BIM in the Czech Republic was legislatively launched by the Resolution of the Government of the Czech Republic No. 958/2016 on the importance of the BIM method for construction practice in the Czech Republic. The Ministry of Industry and Trade was entrusted by the Government of the Czech Republic with the implementation of BIM, which developed through the Czech Agency for Standardization the Concept of BIM implementation in the Czech Republic [1]. In the Ministry of Transport, the guarantor of BIM implementation is the State Fund for Transport Infrastructure, which developed the Prescription for Building Information Modelling (BIM) of Transport Infrastructure [6].

The Railway Administration implemented the introduction of BIM technology into its organization with the document Strategy for the implementation of the BIM process with the assumption that it would play the role of data manager in the BIM process [7]. Currently, the documentation of railway infrastructure is heterogeneous and consists of about 65% of data in digital form and about 35% of data in analogue form. The aim is to unify the availability of data through a common data environment with efficient tools for data management and updating.

BIM data must be highly standardized, especially for the usability of machine-readable exchange formats that are used to update BIM data during the operation phase of the construction. Therefore, it is necessary to emphasize the standardization of the format and the standardization of the data contents.

Format standardization at the international level in the field of BIM is addressed by the IFC (Industry Foundation Classes) standard, which is already implemented in the CSN standards. The IFC format serves as an interchange format between different BIM software.

Contents standardization is about using the so-called LOIN (Level of Information Need). A BIM model contains geometric and non-geometric data and therefore two levels of detail (LOD) are distinguished, namely the Level of Detail of Geometry (LOG) and the Level of Information (LOI).

BIM models must be georeferenced in a binding geodetic coordinate and height reference system because the compatibility with the cadastre data maintained in the Information System of the Cadastre of Real Estate must be ensured due to the property law aspect. In the Czech Republic, this is in the Datum of Uniform Trigonometric Cadastral Network (S-JTSK) and in the Baltic Vertical Datum—After Adjustment (Bpv). The Unified Geodetic Reference System must be correctly implemented in the BIM software. Cartesian coordinate systems in BIM or CAD software are mathematically right-handed, while geodetic systems are mathematically left-handed. Therefore, when the coordinate systems differ, the geometric data must be transformed into the BIM environment. Another specific feature of the BIM environment is the modelling of elevations. The geodetic height reference system and the position coordinate system do not form a Cartesian coordinate system together. Problems arising from this will become apparent in largescale constructions. The designer designs the building objects into a geodetic datum whose dimensions are converted into zero elevation and the corresponding cartographic representation of the geodetic reference system. Particularly in the case of projects for long sections of transport structures, the problem of distortion of dimensions is significantly manifested. Therefore, it is also necessary to take this into account in the BIM environment. The problem of georeferencing a BIM model in IFC format can be found, for example, in [8,9].

### 1.3. Related Works

1.3.1. GIS and BIM Applications in the Civil Engineering Industry

The current database solutions are accessible via a standard web interface, thus ensuring their accessibility for all construction participants. The possibilities of integrating GIS and BIM are discussed in the research paper [10]. A comprehensive, systematic review on data-level BIM/GIS integration from the perspective of information flow is described in [11]. The main outcomes and findings of this review are as follows. (1) A unified framework that incorporates all data processing tasks for BIM/GIS data integration was developed. (2) Most problems regarding BIM-to-GIS data conversion have been solved, which enables application-level BIM/GIS integration, while those unsolved problems are mainly related to representation transformation and semantics mapping. (3) To ensure a fluent information flow, BIM models should be more reliable, the conversion paths should be more robust and efficient, and a more flexible data model on the GIS side is in need. In [12], it is presented systematic investigation into these two technologies, trying to propose the proper one for BIM/GIS data integration In Civil Engineering. The purpose of paper [13] is to provide an insight regarding the information lifecycle during Design & Construction in the Rail Infrastructure project and investigate the impact of current information management processes—and in particular Standards such as IFC,—on BIM-GIS interoperability and lifecycle management of an asset. The study [14] investigates the topic of BIM/GIS integration with the adoption of ontologies and metamodels, providing a critical analysis of the existing literature. Ontologies and metamodels share several similarities and could be combined for potential solutions to address BIM/GIS integration for complex tasks, such as asset management, where heterogeneous sources of data are involved. Algorithms to interpret the commonly used profiles of swept solid for the transformation of IFC into shapefile are developed. A bridge model is used to validate the proposed method, and a Web GIS-based bridge management system is developed to demonstrate a possible usage of the transformed shapefile model [15].

1.3.2. BIM in Terms of Standardization

There are many definitions of the meaning of BIM. Most of them show different explanations depending on the purpose of use. With regard to the international standardization process, the most important definition is the one according to the technical standard EN ISO 19650 [16], which reads as follows:

"Building Information Modelling = the use of a shared digital representation of a built asset (building) to facilitate the design, construction and operational processes to create a reliable basis for decision making". BIM encompasses not only the actual creation and editing of data, but also the processes required in these actions. It is necessary to determine how the data is handled, for what purpose it is created, who creates the data, and who reads and processes it and when.

In the sense of the above definitions, it is necessary to distinguish between BIM as a model or information model of the building, i.e., as a form of information database, and BIM as a modelling process that uses the information model of the building for the exchange and sharing of information. These exchanges and sharing must take place throughout the life cycle of the building and between all participants. Thus, the BIM method requires the establishment of rules of collaboration and their adherence.

In the document Concept for the introduction of the BIM method in the Czech Republic [1], three basic conditions were identified as the closest objectives for the determination of contracts using the BIM method:

- digital information model of the building (in 3D),
- use of CDE and linked systems (e.g., Computer Aided Facility Management—CAFM) for data and information storage,
- creation of special contractual arrangements, the so-called BIM-protocol, including the BIM execution plan (BEP).

The simplified principle of BIM processes is shown in Figure 2. A new drawing or document is created and processed in the Developed phase, then through review and approval it moves to the Shared phase where it is visible to other professions for coordination. After coordination with other project participants, it goes to the Published phase, which means sharing with the investor.

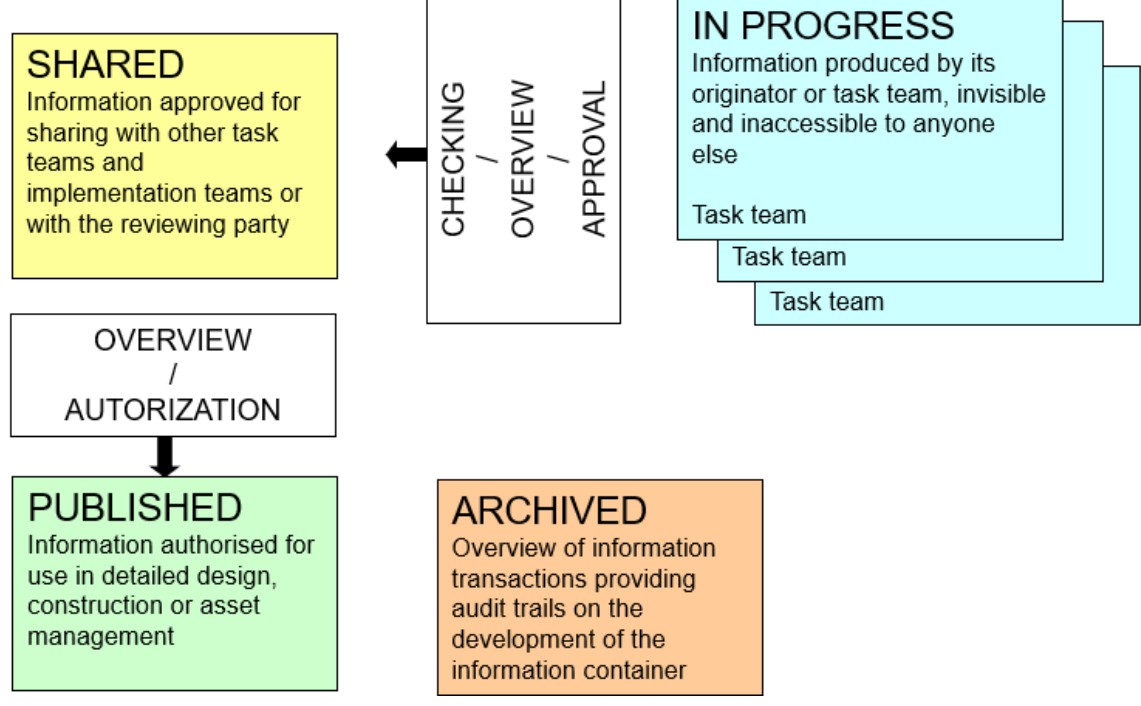

**Figure 2.** States of information containers stored in a common data environment, taken from EN ISO 19650-1 (Figure 10 of this standard).

### 1.3.3. BIM in Transport Infrastructure

In the research study [17], the authors analyze the use of BIM in transport infrastructure in terms of the entire construction life cycle, education, human resources and available literature. Practical experiences and descriptions of typical problems encountered in the implementation of BIM are also valuable. The main issues to be addressed include: handling large data sets, integration of new data unrelated to the object being modelled and data exchange without loss of information. It is proven that the use of BIM in combination with photogrammetry is suitable and highly effective for small projects in the construction industry [18]. Another topic of using BIM is for building monitoring. In paper [19] a human-machine interface is proposed for this activity, in publication [20] the use of IoT and in [21] a special monitoring system is proposed. The case study [22] deals with the issue of CDE (Common Data Environment) and BIM model management in the design and construction of selected large public construction projects. The publication [23] highlights the gap between the general development of BIM and the local, unique context of BIM use. The study also offers potential managers a tool to localize BIM and leverage its benefits for their organizations and sites. The use of BIM as an integration tool in the design of steel buildings is described in [24]. The authors have used previous experience from the literature supplemented by a questionnaire survey. The aim is to improve the project management of steel buildings as defined by critical stakeholders in the steel industry. In addition, a real-time case study is presented to illustrate the contribution of the research. In [25], it is presented a case study of BIM implementation in the management and handover of an underground rail project and tunnel construction [26]. The research presented in [27] investigates the reconstruction projects using statistical analysis, multi-criteria decision-making methods and application of BIM methodology. A case study of

transport infrastructure in the Southern Italy region is presented in [28]. In this study, the parametric modelling method was used in reverse engineering. Another case study [29] focuses on the automation of the production of building elements in the context of BIM. The use of BIM for the facility management phase is discussed in [30]. The linking of Geographic Information Systems (GIS) and BIM in the permitting phase of construction is described in [31]. This is a case study in Norway in which the mapping of GIS information to IFC format coding was investigated. Analysis of the application of the BIM methodology to a case study of the rehabilitation of a railway line using geotextiles and geogrid in a gravel base [32]. The creation of 3D and 4D BIM models was carried out using different BIM-based tools, which allowed to achieve a spatial and parametric representation of the track and to simulate the main design tasks. In the case study [33], the situation of BIM implementation on railways in Morocco is analyzed. It proposes procedures to accelerate the implementation of BIM in railway construction and to approach the developed countries in this field.

### 1.3.4. BIM in Railway Transport

The review study [34] analyses the BIM issues over the last 10 years and focuses on the monitoring of railway structures. The authors propose an extension of the IFC format to capture parameters using sensors, address the optimization of sensor data and interoperability between different BIM platforms. Furthermore, they address the optimization of different sensing technologies for error detection and management of large amounts of data and the consideration of environmental impacts of railway structures. Surveying of railway tunnels using laser scanning and automatic identification of structural elements on the track is discussed in [35]. The method involves two basic operations: point cloud classification and model parameter estimation. The authors of [36] propose a new algorithm to measure the similarity between two BIM components based on their attribute information and similarity. The method enables efficient retrieval of information from BIM models for further use (e.g., SMART City). Testing the geometry quality of railway structures in the context of a BIM model is the topic of paper [37]. The paper details formal methods for checking the correct link between 3D BIM and bill of material (BOM) and describes formal approaches for checking the semantic-geometric coherence of BIM objects. The study [38] is the first in the world to integrate BIM and machine learning to locate defects in railway infrastructure. The developed learning model is beneficial to the railway industry for better asset management and efficient maintenance. A similar issue is addressed in [39].

In [40], Infrastructure Building Information Modelling (InfraBIM), which is a management information system of digital processes for infrastructure, is described. The real innovation lies in the creation of a plug-in that allows extrapolation directly from the design program to the computational model, which then simulates the proposed transport structure in a real context. The project of using BIM in conjunction with the Internet of Things (IoT) is described in [41]. It is an efficient planning and design of rail transport lines with 3D visualization and scenic modeling of the railway environment. The system can be used for visual management of railway infrastructure operation and maintenance. The prevention of risks on the rail line is addressed in [42]. The central element is the LOD (Level of Detail) in the BIM model, which represents the risk object (e.g., a switch) with all geometric and non-geometric information. The main objective of the project in [43] was to systematize the information of the digitized railway infrastructure in order to assess possible performance gaps within the national infrastructure. The project proposed a digitization strategy to define surveying activities and implement openBIM systems to develop an object library and a federated digital asset management model. In [44], the authors developed a 3D BIM model based on the IFC standard, which is extended with geometric information description and hierarchy to model the multi-component subsoil in a railway station. On this basis, an automatic workflow is established to convert the original design data into the corresponding BIM model. Optimization of design in railway

construction using BIM is also addressed in other works [32,45–50]. Test of the possibilities and capabilities of storing GIS data in the open standard Industry Foundation Classes (IFC) for BIM was solved in [51]. The study [52] analyses the requirements specified in the Deutsche Bahn AG guidelines for the technical design of railway structures and investigates the feasibility of implementing these rules in a BIM model. A specific application of BIM to a station building in London using Revit-based simulation of the construction works was discussed [53]. It proposes the transformation of a 3D building model into a 6D building information model. A comparative study of railway building projects in South Korea is contained in [54]. The projects were compared in terms of the use of conventional design using CAD and BIM technology. The results showed that by using BIM, not only the errors in the designs can be detected, but the benefit-cost ratio is increased up to 1.36. In [55], a prototype BIM library based on South Korea's national standards for railway infrastructure is described. Testing showed significant productivity improvements, e.g., an average difference of 38.2% for pavement modeling and an average difference of 50.2% for bridge modeling.

### 1.3.5. Transformation of Existing Railway Construction Documentation from CAD to BIM

Conversion of CAD data to BIM is dealt in [56]. The proposal is based on geometric models and methods for converting geometric objects from AEC to BIM. The research is both theoretical and experimental. Methods of geometric 3D modeling, algorithm and program development, object-oriented architectural modeling, and integrated data analysis and synthesis have been used. A new approach to CAD and BIM data integration based on a single building model that encapsulates both CityGML and IFC models, avoiding translations between models and loss of information is described in [57]. To build the result, all classes and related concepts from both models were first collected, then overlapping concepts were merged to create new indoor and outdoor objects, and finally the spatial relationships between objects were redefined. Unified Modeling Language (UML) notations were used to represent its objects and the relationships between them. The interoperability of 3D GIS and ontology-based BIM (Semantic Web) is dealt in [58]. The conversion between the two models is implemented by CityGML and IndoorGML tools. A new software is proposed that automatically generates BIM models from two-dimensional (2D) CAD drawings [59]. The proposed BIM generation method, contains information about other properties of structural elements and assemblies in addition to 3D models.

## 2. Materials and Methods

The objective of the study is a variant remodelling of existing data on a railway track segment containing a representative sample of typical railway elements based on existing standards and specifications of data models, and evaluation of the achieved results.

In the following part of the text, two variants of remodelling the geodetic part of the as-built documentation (G-ABD) into the BIM environment and alternatively also into the GIS environment will be described. The need to remodel the documentation into BIM is based on the concept [1] and the future legislative obligation in the Czech Republic to tender public procurement with a value of over EUR 6 million using BIM from 2023. The need for remodelling into GIS is based on the legislative requirements of the law [4], which introduces the DTM CZ and the related need of the Railway Administration to provide selected guaranteed data of the railway transport infrastructure into the DTM CZ. The remodeled data were the survey data of the as-built state of the construction of the reconstruction of the railway station Šumice, which is located about 85 km east of Brno in the Czech Republic (Figure 3a). Survey data was provided by the Railway Administration. The structure is located on the single-track national railway line Staré Město u Uherského Hradiště—Vlárský průsmyk. It was about 300 m long section of the railway line, on which the platform, bridge (Figure 3b) and station building were reconstructed (Figure 3c). Along with this, reconstruction of the superstructure and substructure and the cable ducts was also carried out. The authors are aware that this is a relatively short segment of the track, which,

however, provides a necessary sample of elements of the technical railway infrastructure for the remodelling of the documentation.

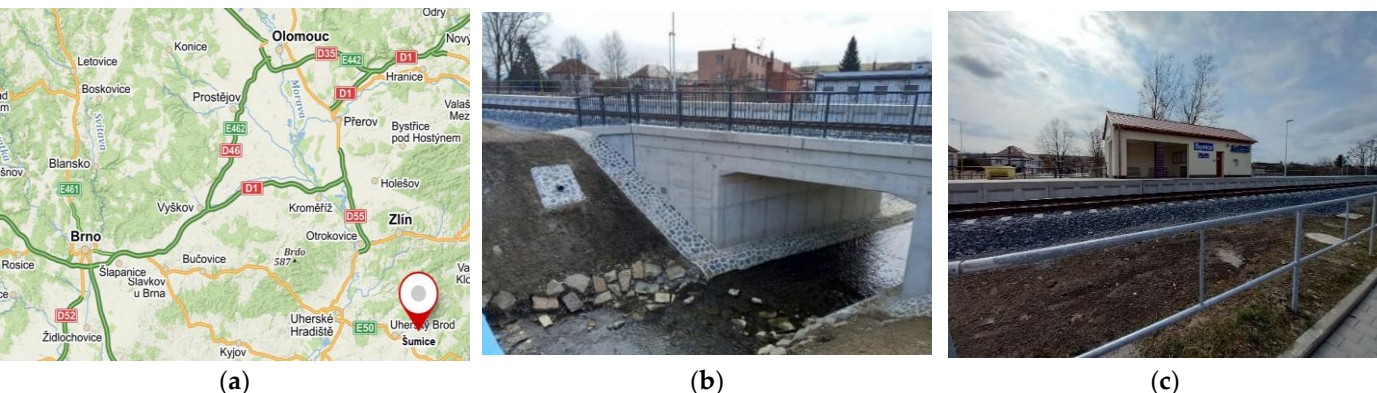

|(**a**)|(**b**)|(**c**)|

**Figure 3.** (**a**) Location of the construction, (**b**) Part of the railway line with a bridge, (**c**) Railway station building.

The graphical output of the current G-ABD, prepared according to the current valid standard of the Railway Administration M20/MP005 [60], is a drawing of the new situation in DGN format. An example of a cut-out of the current G-ABD documentation is shown in Figure 4. It is a 3D vector drawing with 2D elements of point objects (expressed by cartographic symbols) representing special railway objects such as traffic signals, lighting towers, radio, etc. The DGN drawing also includes text descriptions.

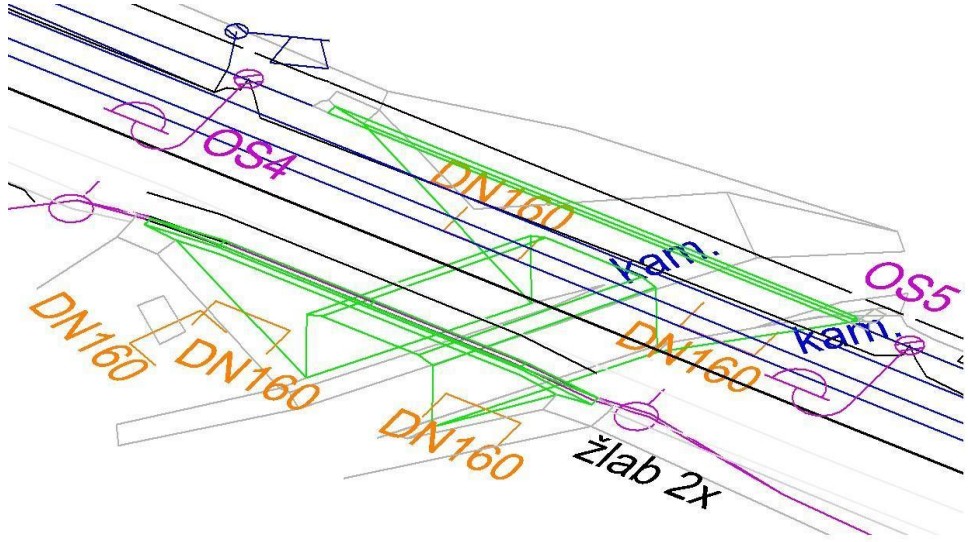

**Figure 4.** Cut-out of the DGN drawing of the graphical part of the G-ABD.

According to [61], the following common data environment platforms are used in the pilot BIM projects of the Railway Administration. The pilot projects are related to the implementation of the BIM approach in the Railway Administration. They were aimed at gaining experience for the creation and validation of BIM process concept documents, which are part of the BIM concept [1]. On the basis of the pilot projects, internal concept documents and requirements for changes to internal processes for the implementation of the BIM method in the Railway Administration organization were developed. Different BIM environment platforms (CDEs) were used in the pilot projects. Table 1 shows for each CDE platform the number of occurrences in each pilot BIM project.

**Table 1.** Occurrence of CDE platforms in RA pilot BIM projects.

| CDE | Number |
|---|---|
| Bentley ProjectWise | 5 |
| Proconom | 1 |
| AspeHub | 1 |
| BIM360 | 1 |

Bentley ProjectWise CDE was used in our case study due to its most common occurrence and also due to the easy availability of the license for our department of the Institute of Geodesy FCE BUT. The tools available in the Bentley ProjectWise CDE can be divided into two services:

1. ProjectWise 365 Services, which are used to manage all project information,
2. iTwin Services—i.e., the digital twin, which is used to operate the basis of BIM, i.e., the 3D model.

An overview of the tools of the Bentley ProjectWise common data environment is given in Table 2.

**Table 2.** Bentley ProjectWise Common Data Environment Tools.

| Tool | Short Overview |
|---|---|
| Share | It is a basic tool used to upload (share) data from individual desktop software (offline) to the CDE environment (online). |
| ProjectWise Web Connection | ProjectWise also exists as a desktop application, this tool in the online CDE is used to connect to the offline version of the application. |
| Deliverables Management | This tool is used to accept documentation. Both the client and the contractor are represented in the CDE and the handover (making available to the other party) takes place via this tool. |
| Project Insights | Here you can find an overview of basic information about the project. |
| Portfolio Insights | Here you can find an overview of basic information about the portfolio (i.e., all projects of the user). |
| Issue Resolution | The BIM model can be commented on. Each participant in the process can be assigned a task to do. These comments may (but need not) be linked to a specific location in the 3D model. |
| Forms | Tool used to create forms that can be used to facilitate the coordination of people. The advantage is the possibility of predefining the content of the form and the impossibility of unauthorized persons to interfere with the template. |
| Components Center | With this tool, it is possible to insert small BIM models of individual building components into the model, which are created by the manufacturer and made available to the contractor. |
| PlantSight | This tool is used for analyses and summaries of model elements. |
| iModel Manager | This tool is used to manage, view and coordinate 3D models. It is a basic tool that allows you to view the geometry of the model. |
| OpenTower IQ | This relatively new tool is used to streamline communication in relation to 5G networks. |
| OpenUtilites Digital Twin Services | This tool is used to visualize utility networks. |
| ProjectWise ContextShare | Tool to manage data storage so that you don't have to copy documents and overwhelm memory. |

### 2.1. Variants of Documentation Remodelling

The principle of converting CAD data into BIM and GIS is shown in Figure 5. The GIS environment in our case consists of geographic objects (GO) of 3 types:

- VGO—a virtual geographical object that does not yet physically exist in the real world, but exists in the form of an idea for a future building.
- RGO-CAD—a real geographic object with CAD documentation, i.e., a 3D geometric model of the object.
- RGO-BIM—a real geographical object with documentation in BIM technology, i.e., an information model of the object.

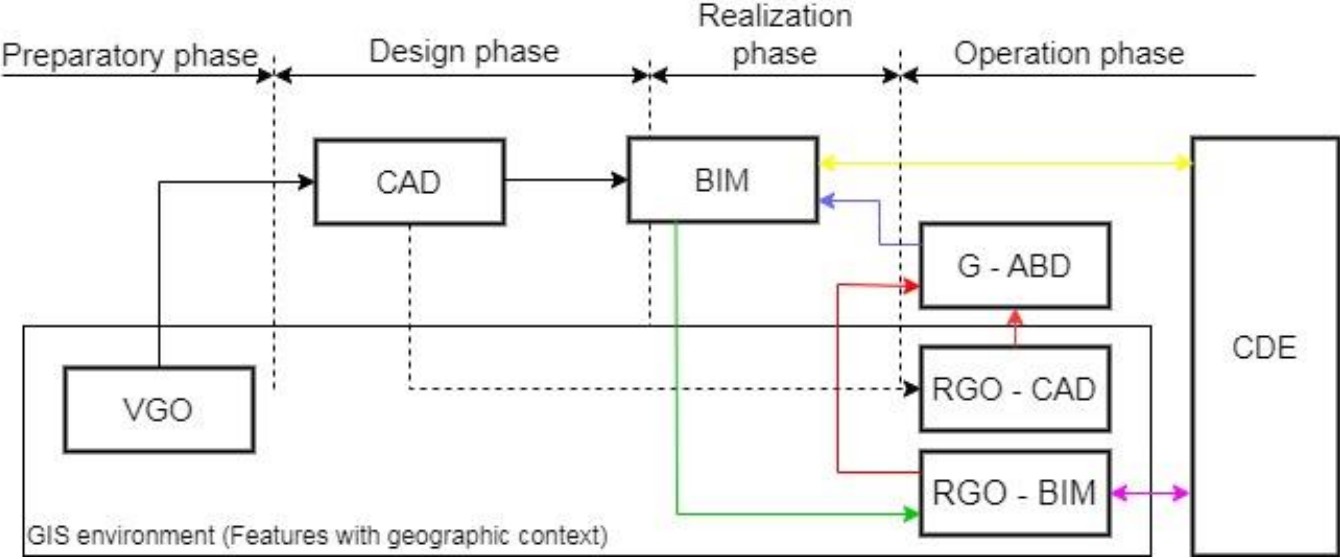

**Figure 5.** GIS—BIM in the context of the construction life cycle. CAD—3D geometric model of the building. BIM—Building Information Model. VGO—Virtual Geographic Object—non-existing feature (idea). RGO-CAD—Real Geographic Object—existing feature in CAD. RGO-BIM—Real Geographic Object—existing feature in BIM. Red line—geodetic surveying of existing structure. Blue line—transformation G-ABD to BIM. Green Line—transformation G-ABD to GIS. Orange Line—object (construction) without geographic context (core of the construction). Violet Line—object (construction) with geographic context (object + environment)—built environment.

Mathematical model of CAD to BIM remodelling

$$(C_P \times C_L \times C_A) \cup C_T \rightarrow \bigcup_{i=1}^{n} B_i \cup B_T \tag{1}$$

where

$C_P$ is set of point elements in CAD
$C_L$ is set of line elements in CAD
$C_A$ is set of area elements in CAD
$C_T$ is set of attributes in CAD
$B_i$ is set of 3D elements in BIM (Level of Development)
$B_T$ is set of attributes of BIM elements.

Equation (1) can be divided into graphical and non-graphical parts.
Graphical part:

$$(C_P \times C_L \times C_A) \rightarrow \bigcup_{i=1}^{n} B_i \tag{2}$$

Non-graphical part:

$$C_T \rightarrow B_T \tag{3}$$

The projection in Equation (2) is a single-valued function, the projection in Equation (3) is not a single-valued function—see Figure 6.

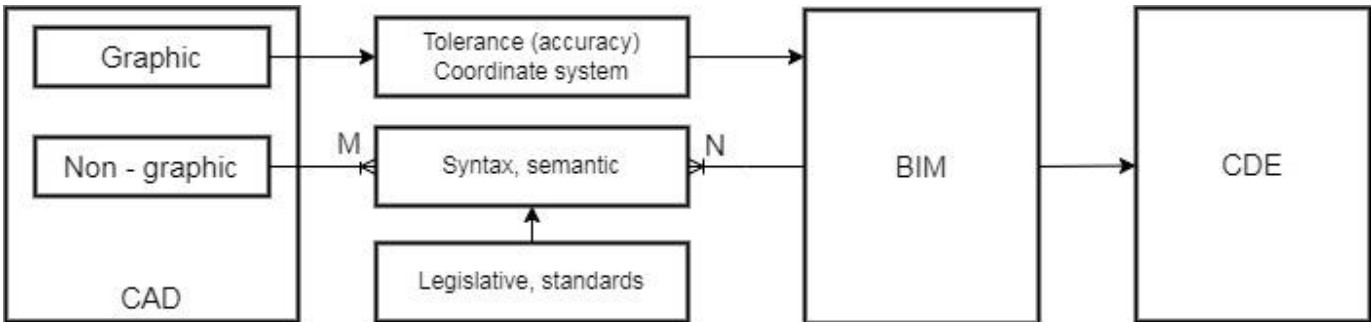

**Figure 6.** Conversion from CAD to BIM.

Mathematical model of CAD to GIS remodelling

$$(C_P \times C_L \times C_A) \cup C_T \rightarrow (G_P \cup G_L \cup G_A) \cup G_T \tag{4}$$

where

symbols $C_P$, $C_L$, $C_A$, $C_T$ have the same meaning as in the Equation (1),
$G_P$ is set of graphical point elements in GIS (point layers)
$G_L$ is set of graphical line elements in GIS (polyline layers)
$G_A$ is set of graphical area elements in GIS (area layers)
$G_T$ is set of element attributes in GIS.

We can also divide Equation (4) into graphical and non-graphical parts.
Graphical part:

$$(C_P \times C_L \times C_A) \rightarrow G_G \tag{5}$$

where
$G_G$ is set of graphical elements in GIS.
Non-graphical part:

$$C_T \rightarrow G_T \tag{6}$$

The projection in Equation (5) is not a single-valued function, the projection in Equation (6) is a single-valued function.

Currently, railway infrastructure constructions in the Czech Republic are not designed using BIM, this approach is only gradually being introduced. On the other hand, there is an existing 3D CAD documentation of the railway infrastructure on the length of the railway tracks of approximately 9600 km, which is subject to remodelling into the necessary newly emerging BIM and GIS platforms.

Figure 5 shows two possible interaction processes of CAD—BIM—GIS technologies in the construction life cycle:

1. The intention to implement a new building (selection of a suitable location of the VGO based on spatial analyses in GIS), design of the building in CAD (3D geometric model) and its supplementation with information in BIM (building information model) including storage in CDE. After the construction is completed, the as-built state of the construction is geodetically surveyed and the information obtained is added to the BIM + stored in the CDE. The next step is the conversion to GIS, which gives the BIM building information model a geographical context.

2. In the case of an existing building with CAD documentation based on a geodetic survey of the as-built state of the construction. The next step is to convert to BIM including saving to CDE. Then the whole model is converted to GIS, giving it a geographical context.

Simplistically we can say that the BIM model itself contains detailed information about the building without a wider geographical context, the object in the GIS environment contains less internal information about the object, but with a real geographical link to the

surroundings, i.e., the geographical context. Thus, the combination of both technologies provides the maximum information content about the internal object and its relation to its surroundings.

The first step of the G-ABD conversion to the BIM and GIS environment was to compare the classification of elements of the data model defined by the RA M20/MP005 standard [59], according to which the G-ABD was prepared, and the data model of the data standard [4], to which the data were to be alternatively transformed, both to the BIM environment and subsequently to the GIS. This section is elaborated in more detail in Chapter 3.

Only those elements of the data model that were present in the subject building were compared in the analysis, not all elements of the data model. The set of elements contained an element from each object category of the RA standard.

### 2.2. Concept of Remodelling CAD Data to BIM Environment

The data transformation scheme from CAD to BIM has two parts (see Figure 6):

1.  CAD vector graphics conversion. Graphical data are geometrically distorted with respect to the used cartographic representation and their accuracy corresponds to the used methods of their acquisition. The current BIM software allows to display objects in Cartesian coordinate system with a possible other possibility of coordinate expression based on transformation between different coordinate systems, which allows to display data obtained by geodetic surveying in field [62].
2.  Conversion of non-graphical information into BIM. The essence of the conversion is the mapping of attributes from CAD to the data model, which is determined by internal legislation (internal standards). This is a difficult process because the cardinality of the corresponding items is generally M:N and both the semantics and syntax of the relevant attributes must be addressed.

All CAD to BIM conversion activities cannot be fully automated. Therefore, it was necessary to use manual remodelling as well, more details in Chapter 3. The CAD drawing is displayed uniformly in the positional component in the cartographic view plane (i.e., with scale distortion) and the third dimension (elevation) is undistorted. In the CAD to BIM remodelling, this principle has been retained for 100% compatibility with non-property cadastre data and related services on top of it.

The simplest and fastest solution for converting CAD drawing content is to import it into the BIM CDE and create a database model for archiving digitized analogue documentation in the form of PDFs, images etc.

Another option is to remodel the CAD vector drawing. The basic step of converting the data into the BIM environment is to remodel it in the BIM software. While CAD drawings store data in layers, a BIM model is an object-oriented database and it is therefore necessary to model 3D objects as 3D solids to which descriptive information will be linked. For the actual creation of the models two software were used. The first one is Autodesk Civil 3D (native DWG format), the second one is Autodesk Revit (native RVT format). Autodesk Civil 3D product is more suitable for the creation of BIM models of linear constructions, while Autodesk Revit product is more suitable for the creation of BIM models of civil constructions. Model creation consists of two phases—creating the geometry and then adding descriptive information. The final step is the import into the Common Data Environment (CDE).

From the point of view of area conversion in BIM, it is necessary to solve that the areas are modelled on the basis of geodetically surveyed points and lines or planes. Thus, in reality, an imperfect straightness or flatness is modeled by a line or plane, which leads to generalization and a slight, sometimes up to several centimeters, distortion of reality. Figure 7 shows a simplified illustration of the principle of generalization of the actual shape compared to the modeled shape and the principle of the origin of deviations between the model and reality.

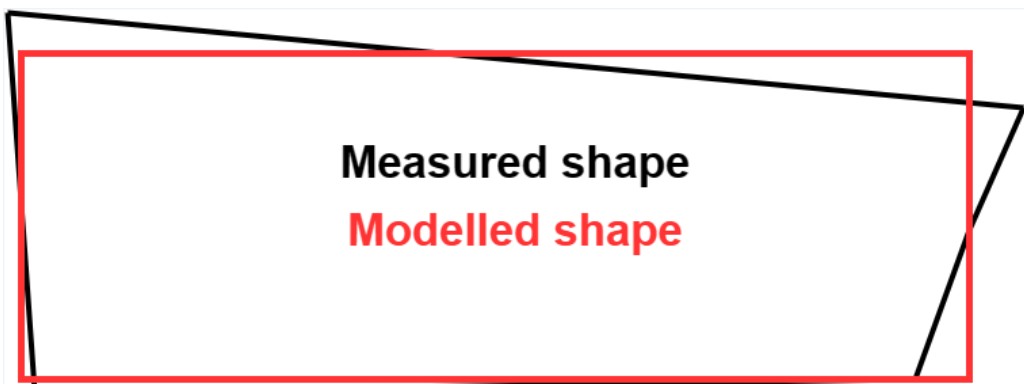

**Figure 7.** Model deviations after generalization (exaggerated).

When remodeling to 3D BIM, the actual oriented shapes of the individual parts of the objects cannot be preserved, because the BIM software only works with perfect geometric parameters of orthogonality, parallelism, verticality. When modelling reality, generalization of shapes necessarily occurs.

The problem of generalization of shapes is also related to the modeling of point objects in the terrain oriented by a characteristic point with the subsequent assignment of a cartographic symbol for its display. For example, a tower is oriented with its centre and height at ground level and depicted by a point cartographic symbol, a railway superstructure is depicted only by its axis, i.e., by a broken line of the spatial position of the track, etc. Therefore, it is not possible to fully model the depicted railway facilities objects from the current CAD documentation in BIM. In the future, according to [60], the Railway Administration envisages the creation of a BIM library of railway equipment objects.

### 2.3. Concept of Remodelling CAD Data to GIS Environment

The transformation scheme from CAD to GIS also has two parts (see Figure 8):

1. Conversion of vector graphics from CAD to GIS, where the problem of displaying (mapping) layers from CAD to GIS layers is solved. This view is not generally compatible. In CAD it is possible to insert elements of different geometry types into the same layer, in GIS each vector layer must have its own geometry (point, line, area). The layer mapping generally has an M:N cardinality—see Figure 8.
2. In the case of converting non-graphical information (attributes), the mapping is 1:1, but a restructuring of the attribute tables is required. Attributes need to be mapped to features according to layers in the GIS.

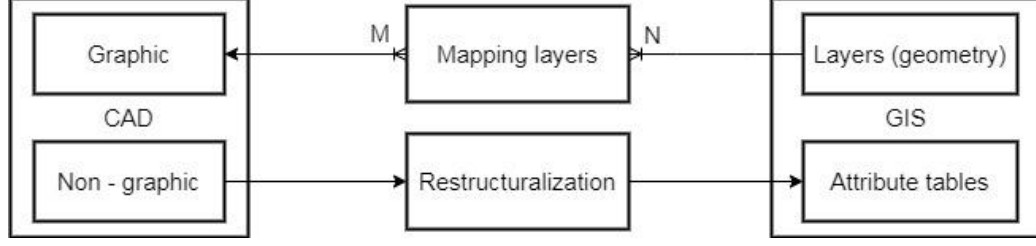

**Figure 8.** Conversion from CAD to GIS.

At this stage, the reasoning was based on the fact that the existing CAD documentation of the railway infrastructure is in 3D, the BIM should be a model in 3D and the DTM CZ should be in GIS in 2D or as 2.5D as a surface description. The basic step of converting data into a GIS environment is its remodeling in GIS software. CAD drawings have data stored in layers, the GIS model is an object-oriented database with 2D geometric objects to which descriptive information can be linked. ESRI's ArcMap software was used to create the

model. The CAD 3D drawing was imported into the ArcMap environment with subsequent redrawing into a 2D plan drawing of the enclosed GIS objects (surfaces). Subsequently, descriptive information was added to the objects in the database.

The first step is to import the DGN into the GIS environment, during which the content of the drawing is sorted according to object types—point, line, area. The goal is to create data content that conforms to the required data standard. Two approaches are possible for creating objects in GIS. The first one is the principle of manual digitization, i.e., the creation of the corresponding objects in layers also with the resolution of the object type (point, line, area). The second approach is data analysis using scripting.

### 2.4. Cartographic Symbols Used for Special Railway Objects

The appearance of the cartographic symbols used by the Railway Administration for creation is defined by the RA M20/MP005 [60]. Special railway objects are usually represented by point cartographic features displayed as 2D objects in a 3D drawing. For standard point and line objects (e.g., shafts, fences, utility routes, etc.), the symbols and line patterns according to the Czech technical standard CSN 01 3411 for large-scale maps are used. Similar issues are addressed in [63]. The work [64] deals with the same topic in the context of GIS.

Cartographic symbols of special railway objects express:

- simplified real shape of the object, e.g., end of platform signaling (Figure 9),
- simple geometric shapes (square, rectangle, triangle, circle) for the superstructure equipment, e.g., axle computer sensor and balise (Figure 10).

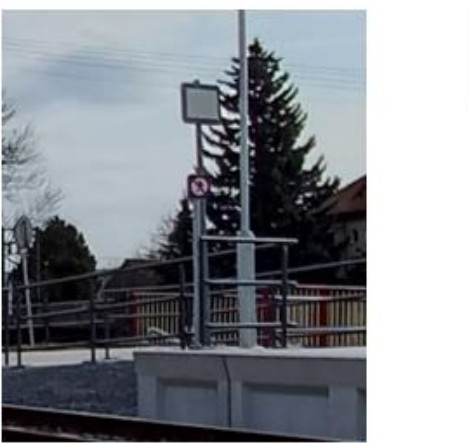
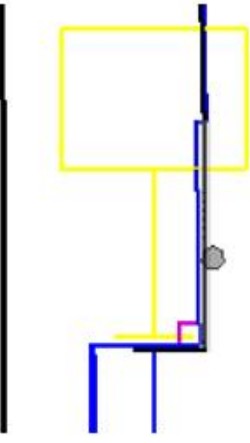

**Figure 9.** End of the platform photo and its cartographic symbol.

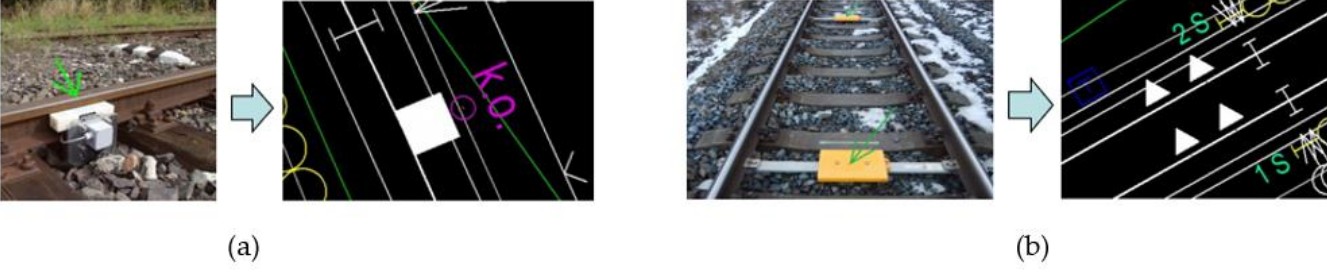

**Figure 10.** Axle computer sensor (**a**) and balise (**b**) and their cartographic symbols [60].

### 3. Results

### 3.1. Remodelling and Conversion of Data into BIM

The left part of Figure 11 shows an example of the original CAD drawing in DGN format of the cable route, in which we can see descriptive information in the form of text

elements. The right part of Figure 11 shows an example of its remodeling in Autodesk Civil 3D software, where the textual data is converted into a database. This data conversion is known, standard and can be automated relatively well.

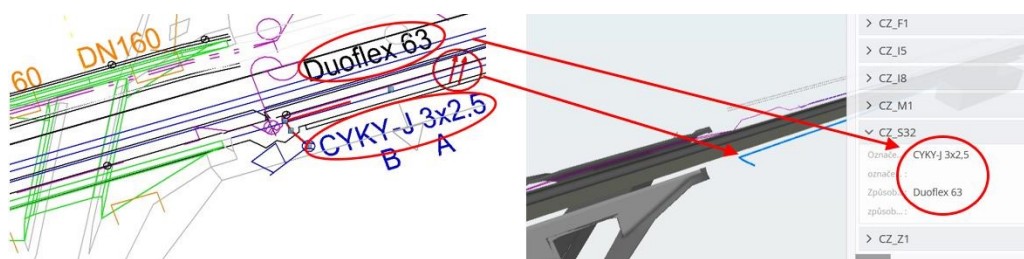

**Figure 11.** Example of CAD to BIM remodeling in Autodesk Civil 3D (lines are transformed into 3D solids and text are transformed into database).

Modelling objects with spatial composition that are part of a linear construction (e.g., bridges) in Autodesk Revit is impractical due to the need to link elements to floors, but advanced modelling tools allow linear spatial objects to be modelled well. The complexity of the shape may necessitate the need to model excess mass, which must then be cut away using the blocks shown in orange in Figure 12.

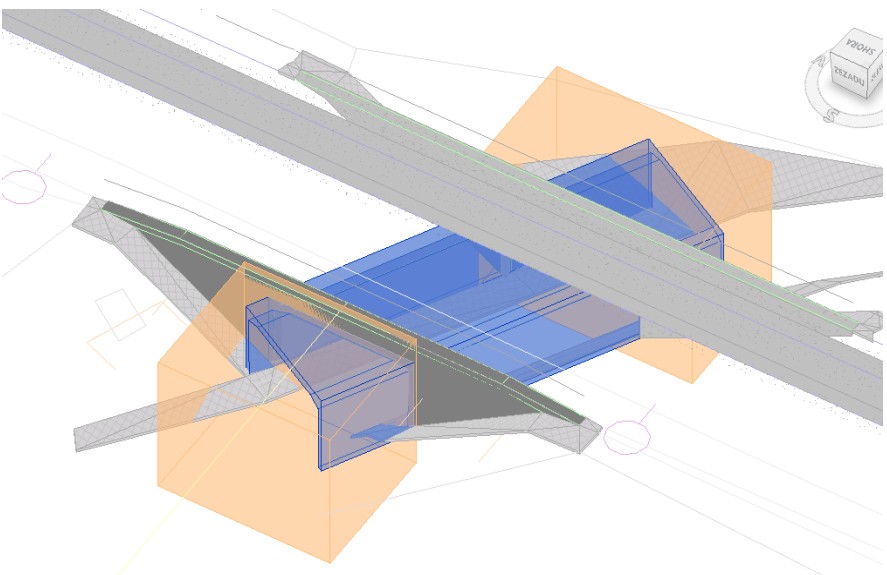

**Figure 12.** Model of a bridge with excess mass (orange) that has to be cut off.

The basis of the CDE is cloud storage and a 3D model viewer that can be used to view descriptive information. Object can be marked, e.g., with a red flag, it is possible to mark elements in the model and comment on them, assign tasks to individuals who can comment on them (Issue Resolution). In this way it is also possible to attach, for example, a PDF file with content to individual elements, which can be, for example, a detailed implementation procedure.

The CAD drawing in DGN format was interactively transformed in Autodesk Revit into a 3D BIM model using parametrically definable tools. Based on the data standard template, the model was supplemented with the required database information obtained from the geometry or imported from external sources. After export to the IFC exchange format, it was possible to convert the BIM model to CDE. The incompatibility of the coordinate systems of the CAD drawing and the Revit environment was solved by the option of using a reference point and a reference orientation. The coordinate system settings

in the CDE were taken from the CAD source drawing. Figure 13a–c illustrate the sub-phases of the process of remodelling the as-built documentation from a CAD drawing by remodeling it into a BIM model and using the IFC format to convert the model into a CDE environment. Bentley ProjectWise CDE was used.

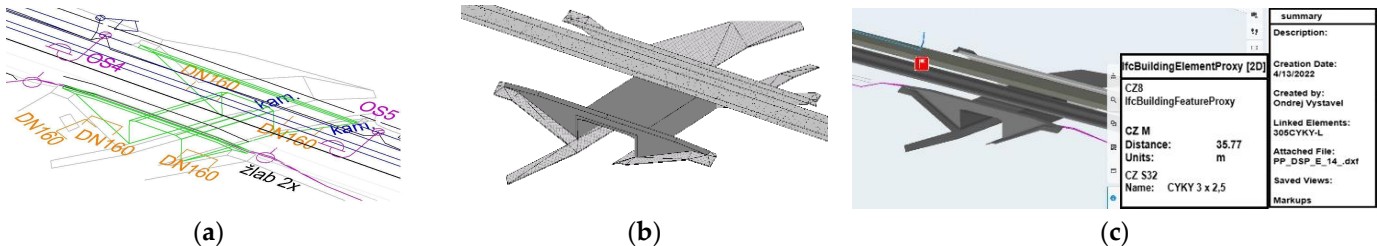

| (**a**) | (**b**) | (**c**) |

**Figure 13.** (**a**) Example of a CAD drawing in DGN vector format. (**b**) Example of remodeling into a 3D model in IFC format. (**c**) Example of import into BIM environment CDE.

### 3.2. Conversion of Data into GIS

The remodelling of our G-ABD data into the GIS environment was done by importing the DGN drawing into the ArcMap software (Figure 14). Topological objects were then made from the drawing and descriptive information was attached to them. The fundamental difference between GIS and BIM is in the detail of the geometric data. A GIS model can be created in 2D and run on standard GIS platforms. The GIS model can also be imported into a BIM CDE.

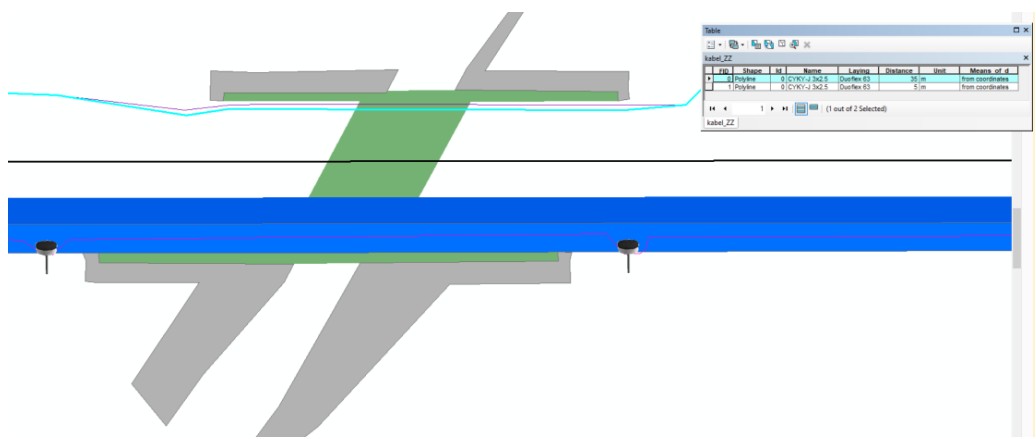

**Figure 14.** Example of remodelling G-ABD data into a 2D GIS model in ArcMap software v.10.

When importing a 3D CAD drawing into a GIS environment, the displayed 2D drawing contains multiple lines corresponding to different height levels, which must be differentiated by attributes. The biggest problem is the subsequent automated creation of closed surfaces. This problem is also solved in the digital technical map of CR (DTM CZ), where the so-called levels are differentiated for all elements. The descriptive information is again attached to the created objects in the form of a database table.

### 3.3. Comparison of the RA Standard [60] and the BIM Standard [6]

The analysis of both standards has shown that the classification of elements in the BIM standard [6] is not in a 1:1 relationship compared to the classification of the RA standard [59]. Only 25% of the elements (e.g., track bed, cable shaft) are convertible in a 1:1 relationship. Most elements are in an N:1 relationship. Thus, several "similar" sub-elements merge into one "parent".

In bridges, the opposite problem occurs, i.e., a 1:N relationship, i.e., one element has to be decomposed into several sub-elements. This is a much bigger problem for CAD to

BIM conversion, as it is not possible to create several different subcategories from one bridge category. A complete statistic of the possible conversion is shown in Table 3. Despite the problems described above, it was possible to classify all the elements, and for the problematic ones a decision process based on experience was followed.

**Table 3.** Comparison of elements of the RA standard and the BIM standard.

| Relationship | Number of Elements | Percent | Example of Element | Description |
|---|---|---|---|---|
| 1 to 1 | 10 | 25 | trackbed | one RA standard element corresponds to one BIM standard element |
| N to 1 | 16 | 40 | culvert | multiple RA standard elements correspond to one BIM standard element |
| 1 to N | 2 | 5 | bridge | one RA standard element corresponds to multiple BIM standard elements |
| N to N | 1 | 2.5 | platform | more RA standard elements correspond to more BIM standard elements |
| | 6 | 15 | shaft | RA standard element can be incorporated into a similar BIM standard element based on experience |
| | 5 | 12.5 | building | RA standard element is addressed by a different regulation |

It is the transfer of data from one structure to a completely different structure. While the RA standard has been in practice and successfully operated for several years, the BIM standard has been formulated recently, it is the first version and is currently being tested on several pilot projects. It appears that the BIM standard will need to be modified in the future.

### 3.4. Comparison of GIS vs. BIM Remodelling Options

The need to rework existing CAD building documentation into BIM or GIS stems from the current needs of large building managers, given that the building was originally designed in CAD and was not previously documented in either a BIM or GIS environment. The fundamental need is to use as much information as possible from the construction documents corrected to the as-built state, including ensuring its effective accessibility, which can be addressed through BIM or GIS.

When comparing the two ways of remodelling CAD documentation into BIM and GIS, it is important to stress that both approaches share a common basis, i.e., the creation of objects to which descriptive information is linked. However, the two approaches differ in the detail of the geometric information. For the purpose of maintenance of the building during its lifetime, it is also advisable to consider the degree of simplification of the documentation, or the possibility of using schematization with appropriate linking to the archive of the execution documentation.

The BIM model is in 3D and thus more detailed, so it detects potential height collisions much better. In GIS, height collisions have to be resolved by analyzing the drawing in the context of attribute-resolved levels, which is less convenient. Another advantage of the BIM model is the possibility of using models of prefabricated parts, which can be easily imported into the BIM model, when modelling building structures.

With regard to the documents used for the design, it must be considered that BIM or GIS models must be compatible in particular with cadastral data and therefore it is necessary to respect the relevant national geodetic reference system, which is defined by

the reference surface and map projection, for their correct geographical location. A specific feature of BIM or GIS models of linear structures is the distortion of the model resulting from the geodetic reference system used.

In terms of georeferencing, this functionality is not completely standard in current BIM software. In the case of the Bentley ProjectWise CDE environment used, the geographic coordinate system parameters of the model can be taken from the initial import of the model data.

In the GIS model, georeferencing is solved quite standardly by assigning the appropriate national geodetic reference system, but it is necessary to change the dimension to a 2D model with database information, or to use tools to support a 3D BIM model placed in the GIS scene. The need to use 2D or 2.5D GIS results from the need for compatibility with the DTM CZ established by legislative regulation [3].

The ESRI ArcGIS Pro platform offers support for accessing BIM data through the Industry Foundation Classes (IFC) format as well as the Revit Project File (RVT) format. This allows you to view all the details, textures and information for individual objects, but without the ability to edit or modify the appearance or structure. The second option is to use the BIM data conversion tool Data Interoperability—Feature Manipulation Engine (FME)—Import data into ArcGIS Pro. The BIM File to Geodatabase tool can be used to convert from IFC or RVT format to ArcGIS Pro file geodatabase. Visualization of BIM objects is handled in ArcGIS Pro through a local scene. Due to the different methodology of object creation in CAD and GIS, it is necessary to consider the specifics of using coordinate systems in CAD so that the BIM model is correctly displayed in ArcGIS Pro. The coordinate systems used in CAD, such as UCS (User Coordinate System) or GCS (Global Coordinate System), are model systems and their relationship to geodetic reference coordinate systems that describe the position of points in the Earth's geospatial domain needs to be ensured. There are two procedures for accurately placing a BIM object in a local ArcGIS Pro scene.

In terms of georeferencing of BIM models, it is necessary to distinguish whether the BIM model is designed 1:1 due to the need for an undistorted dimensional shape and the need for high demands on the realisation of geometric accuracy, or whether the BIM model takes into account the distortion of the respective cartographic represent map projection used (the coordinate system used) with respect to the context and continuity of the spatial position and continuity of long linear objects. There are two basic options for coordinate correct positioning of the BIM model. The first is to user-define a reference point that is uniquely and easily identifiable on an existing BIM model of the object, and assign that point a corresponding geographic coordinate and an appropriate azimuth of rotation for the model so that the model is correctly positioned and oriented in the geodetic reference system. The second option is to use the Building—Georeferencing tool.

In terms of accessibility of the original construction realization documentation, in both variants of processing into the BIM or GIS environment, it is possible to store (archive) files of the original construction realization documentation, e.g., in the form of a database-stored PDF file. It is thus possible to trace the current version of the documentation. The need for using the PDF format results from the need to archive the original tender design (in analogue form) of the construction, which was not maintained in BIM.

If the construction has not been managed in the BIM environment from the beginning, it is more effective to create a simplified model compatible with the DTM and supplement it with the digitized data from the original analogue tender design.

In the BIM CDE it is possible to comment on the documentation (Issue Resolution and Forms tools). The strength of these tools is the identification of the participants. The actions performed are thus unambiguous and archived. In the GIS environment, it is possible to work in a server environment that allows setting access permissions to sub-agendas.

## 4. Discussion

From the point of view of the tools of the Bentley ProjectWise common data environment, the most effective ones appear to be those that enable communication between the

participants in the process. For this, the tools Delivery Management, Defect Resolution and Forms are suitable. In addition, the Component Centre tool—i.e., a library of small BIM models of building components—is very useful. The use of Bentley ProjectWise tools is also evidenced in Table 1 by the higher frequency of independently produced pilot projects [61].

BIM models for use in a common data environment must be georeferenced, otherwise the coordination of multiple models that is assumed in CDE is impossible. The Bentley ProjectWise common data environment does not allow the setting of a coordinate system, it is only possible to take the coordinate system from an imported BIM model produced in desktop software. However, georeferencing brings with it the questions posed in [8,62]. In particular, this concerns linear structures and possibly large objects with spatial composition, where the scale change of the dimension in the terrain and in the plane of the map projection is already apparent.

The necessity of correct georeferencing of BIM models in CDE is caused in particular by the need for interconnectivity with cadastral data and mapping services of technical infrastructure administrators maintained in the national territory-wide binding geodetic reference system. These data are subject to continuous updating and their connectivity to BIM needs to be ensured. The degree of homogeneity of the data is ensured by its accuracy, which can be expressed by an attribute according to which the data can be sorted and analyzed.

In terms of 2D or 3D data dimension, it would be ideal to have all geometric data in 3D, but most of the existing data is in 2D (cadastral data, DTM data, orthophotos, GIS data). Therefore, it is necessary to address the problem of appropriate interconnectivity, convertibility and usability of 2D and 3D data. In the 3D model, the dimensions are expressed in the plane of the used cartographic representation, i.e., with distortion, with respect to connectivity to other data sources.

In the DTM maintained in the Czech Republic in a 2D GIS environment, the expression of height levels is provided in the drawing by its attributes expressing the respective height levels. One of the solutions for the conversion from 3D to 2D is the possibility of converting the IFC format to an SHP file similar to [14,15]. In doing so, the problem discussed e.g., in [57] concerning the loss of part of the information cannot be avoided.

There are several options for remodelling existing CAD documentation into a common data environment. The simplest option is to import 3D CAD drawing data (e.g., in DGN format) into a common data environment. The advantage is the speed of the process, but the disadvantage is the absence of 3D model objects (3D masses), the geometry is represented only by lines. The advantage of a unified BIM environment in this case is mainly in the digital archiving. If emphasis is also placed on the existence of 3D masses in the model, it is necessary to remodel such objects from the CAD drawing and then convert them to BIM. However, the remodeling of objects is time-consuming. It is therefore essential to find the optimum level of detail (granularity) of the model, which must be based on the purpose for which the digital model is to be used and also consider the costs that need to be incurred to rework the model. An option is to remodel only the most important objects.

It is important to determine the needs of the building operator and the resulting need for a simplified model that can be administered by the BIM common environment tool. The main advantage of converting to BIM is the possibility of archiving the original implementation documentation and its availability.

It has proven inefficient to retrospectively create BIM models in as much detail and granularity as if it were the as-built (WDD) stage. It is more advantageous to create a simplified BIM model from the geodetic part of the as-built documentation (G-ABD) and add the as-built drawing data files to it and create the necessary links. In the case of BIM construction, the actual construction documents can be used to verify what was actually built.

Updates to BIM model data during the construction life cycle process can be efficiently performed by importing it into the CDE via the IFC exchange format. The advantage of CDE, and thus the benefit of the whole BIM, is the 3D spatial dimension of the model, the

possibility of archiving the construction documentation and thus ensuring its availability throughout the construction life cycle.

Alternatively, conventional 3D CAD documentation can be reworked into a GIS environment that also supports the incorporation of BIM models into the scene. Compared to BIM, a GIS model has less geometric information due to its 2D dimension, database archiving and accessibility of the construction documentation is possible and the advantage of GIS is e.g., the possibility of connecting publicly available external web mapping services e.g., with cadastral data, data of underground technical infrastructure managers, data of the digital technical map of the Czech Republic, etc. Public geodata are currently maintained on GIS platforms, but even these platforms support work with BIM models.

## 5. Conclusions

At present, the realization phase of railway constructions ends with the preparation of 3D CAD as-built documentation, which is used for its subsequent operation and maintenance. The transformation into a common BIM or GIS data environment is intended to streamline the use of this documentation in the operational phase of the construction, while the BIM model is to be used effectively for e.g., maintenance planning.

In the paper, the method of transformation of data obtained from the as-built state survey into the BIM and GIS environment was presented. The proposed solution was verified on a case study of the realized project of the railway station Šumice (Czech Republic). The BIM data standard created by the State Fund of Transport Infrastructure was used for the documentation remodelling. The research published so far has mainly dealt with the transformation of data from CAD to BIM or GIS, i.e., from a discrete space (CAD) to another discrete space (BIM, GIS)—see e.g., [4,59]. The contribution of the proposed method is the transformation from the real space of the construction to the discrete space of CAD (as-built documentation) and then to BIM and GIS. The conversion identified the following main issues:

- conversion of features and attributes from CAD to BIM/GIS,
- differences in geometric accuracy between source and target graphics of these platforms.

The authors propose an acceptable solution within existing CAD/BIM/GIS applications. It is shown that current software systems are not yet able to capture geometric shapes in real space with 100% accuracy. This fact was confirmed by the results in [65]. Further inaccuracies may result from the exchange formats used, as confirmed by the experience of [57].

BIM documentation is particularly effective when the entire construction process is managed in BIM mode from the start. For existing buildings that have not been managed in BIM from the start, the as-built documentation needs to be reworked into a common BIM data environment. The conversion of existing conventional construction documents into BIM is very time and cost consuming, mainly due to the differences in data structures. From the current state of the RA standard [60] and the BIM standard [6], it has been analyzed that the remodelling of classical 3D CAD documentation into a BIM CDE can be realized automatically from 65%, the rest of the data content is necessary, i.e., 35%, requires the necessity of individual human decision making during the remodeling. This fact is due to the lack of pan-European standards for describing the railway network at the level of the interchange format (e.g., IFC). Therefore, so far non-standardized processes are the subject of research activities not only in the Czech Republic [52,55].

A significant benefit of reworking the original CAD documentation into BIM is the streamlining of approval processes during the preparation and execution of constructions. The most detailed data is generated in the construction realization processes administered in BIM, which then serves as a source of data for passport-type agendas, into which they are taken automated by engineering processes.

Another benefit is the streamlining of data updating of internal systems of the Railway Infrastructure Administration, e.g., Digital Technical Map of Railways, Technical Passport

of Infrastructure and the possibility of data sharing to downstream systems of the state administration (DTM CZ) in accordance with INSPIRE.

The research will continue with the design of a suitable data model that would eliminate these data conversion problems between CAD/BIM/GIS platforms to the maximum extent possible.

The main contribution of the research is the inter-convertibility of data between the completely different data models specified by the Railway Administration standards and the newly emerging BIM standards. The mutual model incompatibility of about 35% and the resulting necessity of modification of existing or design of completely new engineering procedures as well as proposals for changes in some existing organizational and technical regulations of the Railway Administration of the Czech Republic was shown. This results in the preference for the use of modern mass data collection technologies based on laser scanning and surface photogrammetry instead of less efficient remodelling of existing heterogeneous data. On the contrary, remodelling of underground technical infrastructure objects proved to be more efficient.

**Author Contributions:** Methodology, D.B., J.B. and O.V.; data curation, R.H. and O.V.; writing—original draft preparation, writing—review and editing, D.B., J.B. and O.V.; visualization, O.V.; project administration, supervision, R.H. All authors have read and agreed to the published version of the manuscript.

**Funding:** Financial support: Brno University of Technology, specific research project Nr. FAST-J-23-8374 "Development of a BIM Common Data Environment (CDE) model for educational purposes". This article was written with the support of Brno University of Technology specific research project Nr. FAST-S-22-8035 "Research on the use of GIS and BIM technologies in the processes of digital-ization of the life cycle of buildings".

**Institutional Review Board Statement:** Not applicable.

**Informed Consent Statement:** Not applicable.

**Data Availability Statement:** Not applicable.

**Conflicts of Interest:** The authors declare no conflict of interest.

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
