# Peer review of "Case Study of Remodelling the As-Built Documentation of a Railway Construction into the BIM and GIS Environment"

_applsci, doi:10.3390/app13095591_

Round 1
Reviewer 1 Report
This mainly talks about the case, the railway construction completion document transformation into BIM and GIS environment case study. The aim of the case study was to evaluate the effectiveness of reworking the existing as-built documentation of a railway construction into a common BIM data environment, in which further subsequent construction agenda should be managed for the remaining period of its life cycle. What is the contribution of the research? It is not clear. The manuscript seems to be a final report of a project, but not an academic paper. What is the difference between your research and other related work? It's not clear whether you're integrating new engineering processes to make it more efficient or whether you're innovating new technologies to make it more efficient.
Author Response
Cover Letter 1
Dalibor Bartonek*, Jiri Bures, Ondrej Vystavel, and Radomir Havlicek
bartonek.d@fce.vutbr.cz
Institute of Geodesy, Faculty of Civil Engineering, Brno University of Technology, Czech Republic
Dear editorial board of the Applied Sciences,
We are pleased to resubmit our paper „Case study of remodelling the as-built documentation of a railway construction into the BIM and GIS environment ".
Our paper has been previously submitted to this issue, and received an encouragement to resubmit. In what follows, we first give, in short, the main big changes conducted in the manuscript, and then details the exact changes for each of the 2 reviewers.
The authors thank the reviewer for detailed revision of the paper and for valuable comments. The entire text has been revised, changes are highlighted in yellow color.
Comments and Suggestions for Authors - Reviewer #1:
This mainly talks about the case, the railway construction completion document transformation into BIM and GIS environment case study. The aim of the case study was to evaluate the effectiveness of reworking the existing as-built documentation of a railway construction into a common BIM data environment, in which further subsequent construction agenda should be managed for the remaining period of its life cycle. What is the contribution of the research? It is not clear. The manuscript seems to be a final report of a project, but not an academic paper. What is the difference between your research and other related work? It's not clear whether you're integrating new engineering processes to make it more efficient or whether you're innovating new technologies to make it more efficient.
Response:
The entire paper was restructured according to comments and many ambiguities were explained.
The aim of our research is to analyze and innovate existing engineering procedures with the aim of 1. remodelling existing CAD documentation into BIM for the purpose of public procurement, 2. the ability to provide guaranteed data to the IS of the Digital Map of Public Administration and 3. the ability to provide data for the design of new railway structures or their reconstruction. The problem is described in detail in section 1.
The contribution of the research is specified in more detail in the final section Conclusions.

Reviewer 2 Report
The presented paper deals with a case study for the remodelling of drawings into BIM and GIS.
The topic is timely and significant. The practical importance of the paper is high. However, some parts should be presented more clearly. Below, you can find some suggestions that, in my opinion, will help improve the manuscript.
In my opinion, the title of the paper has to be changed to clarify that the described procedure is applicable in the Czech Republic.
I suggest using “remodelling” instead of “reworking” in the whole paper.
Line 16 – BIM is Building Information Modeling, not Management
Line 35 – first, Building Information Modeling has to be written and then add to the brackets the abbreviation BIM.
Line 56 – Please, rephrase the sentence.
Line 76 – Please, explain the accuracy in 2-3 sentences.
In this part of the paper, some references are missing regarding the Dig. Tech map. Legislation, standards, etc.
Line 115 – I suggest using legislation instead of general regulations and standards instead of special regulations.
Line 126 – just building documentation – do not use engineering graphics.
Line 133 – clarify that these are maps
Line 138 – what accuracy are we talking about? Can you quantify it?
In this part of the paper… In the operational phase, often deformation measurement has to be performed. In that case, it is not enough the accuracy of the large-scale mapping.
Line 140 – Clarify that the abbreviation TD stands for Tender Documentation
Line 145 – 206 add references, legislation, standards, etc. The ABD is the core of the paper, I suggest to reduce the other parts and describe the ABD in more detail.
Section 1.2.1. – the references are outdated. Maybe add here some examples from GIS and Infrastructure
Line 224 – The EN ISO standard is missing from the references
Line 240 – what is the abbreviation DiMS stand for? It is from Czech?, the same – CAMF
Line 239 – BEP is abbreviation of BIM Execution Plan, not implementation. The “special contractual arrangements” are also defined in EIR (exchange information requirements)
Line 252-256 this part does not follow the previous text in any way
Line 309 Please explain the abbreviation BOM
Line 314 – superstructure of the BIM model? Do you mean the superstructure of, e.g., a bridge in BIM model?
Line – 315 I suggest using InfraBIM instead of I-BIM
Section 1.2.6 – maybe replace it and add after the part with the description of the building process in the Czech rep.
Line 385 – 395 – It is hard to follow what you want to say; why is it important to notice here?
Line 396 – 411 This is a very important and huge task. This part should be described in more detail if it is possible. Maybe how it is solved in your country, what are the possible strategies, etc.
I suggest condensing the whole 1 and 2 sections. Because it contains a general description of processes and is in its current form, it looks like a review, not a scientific paper. These 2 sections are half of the paper.
Afterwards, please, highlight the contribution of your paper and how it is organized.
Line 413 – explain in 1-2 sentences the 2 variants
Line – 419 a longer part would be better because for so short a railway the difficulties from lines 396-411 are not present (real worlds shape, earths curvature, reference surfaces, etc.)
Why is 2.1 a separate section?
Line 432 – there is no info about pilot projects in the previous text
Line 463 – I suggest not to use reverse engineering as a synonym of surveying measurements.
Line 471 – 479 – both variants are the same in terms of remodelling to BIM. In final, you model an existing building. So it is a kind of documentation creation variant, not a remodelling variant.
Line 490 – explain SFTI
Lines 493-495 – please describe this in more detail and elaborate on the results.
Line 519 -520 So you used manual remodelling?
Also, describe in more detail the problems already addressed – accuracy, earth curvature, georeferencing, etc.
Line 537 – why 2D , why not in 3D, explain it please
Line 540-558 – It is BIM-GIS not CAD-GIS
Line 556 – the procedure of georeferencing by a point and azimuth will not work for longer infra projects. The BIM is created in a plane 1:1 without map projection… Add some advice or proposal, if it is possible. also, describe the Georeferencing tool, if it is possible.
The section results should contain the results.
Please use conversion instead of transformation.
Line 579 – in BIM we are using solids, always no lines and surfaces.
Line 583-585- please rephrase the sentence.
Please explain the generalization in more detail with some results and examples. I think this is belonging to the previous part with the description of modelling.
Fig 11 Try to use a better example.
Lines 597-601 – the procedure you described is CSG – constructive solid geometry, and it is well known
Section 3.2 – why conversion to…? do you really need to convert the data? Do you just import them in IFC, or not?
Line 606 – using a red flag – I guess this is software dependent… not applicable in general…
Line 610-613 – this description of the procedure has to be extended in more detail. And add this to the part about modelling.
Section 3.3 – for the word transformation - the same as above
Lines- 615-625 – this a description of modelling not the results
Line 623 what I the abbreviation DTM CR stand for?
Line 673-674 – why did you not use 3D GIS? Why 2D, it is because of some legislation? Or why? Explain it this or in the introduction part, please.
Line 694 – PDF files. Can you describe how it is planned in the future? Can you formulate some advice or suggestions? It will be very valuable.
Author Response
Cover Letter 1
Dalibor Bartonek*, Jiri Bures, Ondrej Vystavel, and Radomir Havlicek
bartonek.d@fce.vutbr.cz
Institute of Geodesy, Faculty of Civil Engineering, Brno University of Technology, Czech Republic
Dear editorial board of the Applied Sciences,
We are pleased to resubmit our paper „Case study of remodelling the as-built documentation of a railway construction into the BIM and GIS environment ".
Our paper has been previously submitted to this issue, and received an encouragement to resubmit. In what follows, we first give, in short, the main big changes conducted in the manuscript, and then details the exact changes for each of the 2 reviewers.
The authors thank the reviewer for detailed revision of the paper and for valuable comments. The entire text has been revised, changes are highlighted in yellow color.
Comments and Suggestions for Authors - Reviewer #2:
The presented paper deals with a case study for the remodelling of drawings into BIM and GIS.
The topic is timely and significant. The practical importance of the paper is high. However, some parts should be presented more clearly. Below, you can find some suggestions that, in my opinion, will help improve the manuscript.
In my opinion, the title of the paper has to be changed to clarify that the described procedure is applicable in the Czech Republic.
Response to the comments:
I suggest using “remodelling” instead of “reworking” in the whole paper. – the term “remodelling” instead of “reworking” has been changed throughout the text.
Line 16 – BIM is Building Information Modeling, not Management - corrected
Line 35 – first, Building Information Modeling has to be written and then add to the brackets the abbreviation BIM. – change accepted
Line 56 – Please, rephrase the sentence - the sentence has been rephrased.
Line 76 – Please, explain the accuracy in 2-3 sentences. – term “accuracy” was explained.
In this part of the paper, some references are missing regarding the Dig. Tech map. Legislation, standards, etc. - the literature was supplemented.
Line 115 – I suggest using legislation instead of general regulations and standards instead of special regulations. – Added reference to the Building Act [3].
Line 126 – just building documentation – do not use engineering graphics. The phrase “engineering graphic” was removed.
Line 133 – clarify that these are maps – the term “maps” was specified.
Line 138 – what accuracy are we talking about? Can you quantify it? accuracy has been specified.
In this part of the paper… In the operational phase, often deformation measurement has to be performed. In that case, it is not enough the accuracy of the large-scale mapping. - The accuracy was explained.
Line 140 – Clarify that the abbreviation TD stands for Tender Documentation - the TD abbreviation has been explained.
Line 145 – 206 add references, legislation, standards, etc. The ABD is the core of the paper, I suggest to reduce the other parts and describe the ABD in more detail. – the relevant reference was added into reference list.
Section 1.2.1. – the references are outdated. Maybe add here some examples from GIS and Infrastructure. All these references were references have been replaced with current ones.
Line 224 – The EN ISO standard is missing from the references – the EN ISO standard was added into references [2].
Line 240 – what is the abbreviation DiMS stand for? It is from Czech?, the same – CAMF – the abbreviation has been explained.
Line 239 – BEP is abbreviation of BIM Execution Plan, not implementation. The “special contractual arrangements” are also defined in EIR (exchange information requirements) – the comment was accepted.
Line 252-256 this part does not follow the previous text in any way – this text was removed.
Line 309 Please explain the abbreviation BOM – the term was explained.
Line 314 – superstructure of the BIM model? Do you mean the superstructure of, e.g., a bridge in BIM model? – the sentence was rephrased.
Line – 315 I suggest using InfraBIM instead of I-BIM – accepted.
Section 1.2.6 – maybe replace it and add after the part with the description of the building process in the Czech rep. – the entire paragraph has been moved into section 1 above.
Line 385 – 395 – It is hard to follow what you want to say; why is it important to notice here? - this part has been reformulated and supplemented.
Line 396 – 411 This is a very important and huge task. This part should be described in more detail if it is possible. Maybe how it is solved in your country, what are the possible strategies, etc. - the authors tried to describe the problem in more detail.
I suggest condensing the whole 1 and 2 sections. Because it contains a general description of processes and is in its current form, it looks like a review, not a scientific paper. These 2 sections are half of the paper. – The sections 1 and 2 were reformulated.
Afterwards, please, highlight the contribution of your paper and how it is organized. – the goal of the research was specified.
Line 413 – explain in 1-2 sentences the 2 variants – explained in the text.
Line – 419 a longer part would be better because for so short a railway the difficulties from lines 396-411 are not present (real worlds shape, earths curvature, reference surfaces, etc.) – problem was described and explained.
Why is 2.1 a separate section? – the subtitle was removed.
Line 432 – there is no info about pilot projects in the previous text - supplemented and explained.
Line 463 – I suggest not to use reverse engineering as a synonym of surveying measurements. – corrected.
Line 471 – 479 – both variants are the same in terms of remodelling to BIM. In final, you model an existing building. So it is a kind of documentation creation variant, not a remodelling variant. - explained before section 1.2.
Line 490 – explain SFTI – abbreviation clarified.
Lines 493-495 – please describe this in more detail and elaborate on the results. – described in section 2.
Line 519 -520 So you used manual remodelling? – explained in the text.
Also, describe in more detail the problems already addressed – accuracy, earth curvature, georeferencing, etc. ? – explained in the text.
Line 537 – why 2D , why not in 3D, explain it please ? – explained in the text.
Line 540-558 – It is BIM-GIS not CAD-GIS - ? – explained, the paragraph was moved into section 3.
Line 556 – the procedure of georeferencing by a point and azimuth will not work for longer infra projects. The BIM is created in a plane 1:1 without map projection… Add some advice or proposal, if it is possible. also, describe the Georeferencing tool, if it is possible. supplemented and clarified.
The section results should contain the results. – this part was supplemented.
Please use conversion instead of transformation. – corrected.
Line 579 – in BIM we are using solids, always no lines and surfaces. – this problem was clarified.
Line 583-585- please rephrase the sentence. – the sentence was explained.
Please explain the generalization in more detail with some results and examples. I think this is belonging to the previous part with the description of modelling. - the paragraph was moved to the section 2, problem clarified.
Fig 11 Try to use a better example. – the Figure was improved.
Lines 597-601 – the procedure you described is CSG – constructive solid geometry, and it is well known - supplemented by a note in the text.
Section 3.2 – why conversion to…? do you really need to convert the data? Do you just import them in IFC, or not? – corrected in the text.
Line 606 – using a red flag – I guess this is software dependent… not applicable in general…- the text has been reformulated.
Line 610-613 – this description of the procedure has to be extended in more detail. And add this to the part about modelling. - the procedure was described in more detail.
Section 3.3 – for the word transformation - the same as above – corrected – “conversion”.
Lines- 615-625 – this a description of modelling not the results – this paragraph was moved into section 2 Methods.
Line 623 what I the abbreviation DTM CR stand for? - the shortcut has been explained.
Line 673-674 – why did you not use 3D GIS? Why 2D, it is because of some legislation? Or why? Explain it this or in the introduction part, please. - the problem was explained - relation to the digital technical map.
Line 694 – PDF files. Can you describe how it is planned in the future? Can you formulate some advice or suggestions? It will be very valuable. - explained in the final section.
On behalf of all authors: Dalibor Bartoněk
21 Feb. 2023

Reviewer 3 Report
The BIM&GIS integration is an important topic. However, several areas require improvement:
1. The abstract lacks clarity and should be streamlined. It is recommended to provide more context regarding the research in this paper.
2. The Introduction section does not clearly present the background and research status. The frequent use of phrases such as "The paper" and "The work" is unclear and appears as a simple list. It is advised to enhance the presentation, organize the overall logic, and develop a more coherent argument.
3. The images throughout the paper could be more visually appealing and should be of higher clarity.
4. There might be spelling errors in the words "Renodelling" on line 714. It is recommended to thoroughly check for such basic textual errors to enhance the paper's rigour.
5. Please clearly indicate the source of the data used in the text.
6. The Discussion section should better integrate previous studies, ideally by explicitly stating how this research has advanced compared to prior work.
7. The paper's writing style requires improvement, as some expressions are overly colloquial.
Author Response
Dalibor Bartonek*, Jiri Bures, Ondrej Vystavel, and Radomir Havlicek
*) bartonek.d@fce.vutbr.cz
Institute of Geodesy, Faculty of Civil Engineering, Brno University of Technology, Czech Republic
Dear editorial board of the Applied Sciences,
We are pleased to resubmit our paper „Case study of remodelling the as-built documentation of a railway construction into the BIM and GIS environment ".
Our paper has been previously submitted to this issue, and received an encouragement to resubmit. In what follows, we first give, in short, the main big changes conducted in the manuscript, and then details the exact changes for each of the 3 reviewers.
The authors thank the reviewer for detailed revision of the paper and for valuable comments. The entire text has been revised, changes are highlighted in yellow color.
Response to the academic Editor Notes:
- The abstract lacks clarity and should be streamlined. It is recommended to provide more context regarding the research in this paper.
Response:
The abstract was supplemented in such a way that the continuity with the subject of the research was obvious.
- The Introduction section does not clearly present the background and research status. The frequent use of phrases such as "The paper" and "The work" is unclear and appears as a simple list. It is advised to enhance the presentation, organize the overall logic, and develop a more coherent argument.
Response:
Section 1 Introduction has been partially modified and partially supplemented. Additional links have been added and the influence of related works on the issue addressed has been clarified.
- The images throughout the paper could be more visually appealing and should be of higher clarity.
Response:
The images have been supplemented with a description that clarifies their content and increases their readability.
- There might be spelling errors in the words "Renodelling" on line 714. It is recommended to thoroughly check for such basic textual errors to enhance the paper's rigour.
Response:
The entire text has been checked and any typos removed.
- Please clearly indicate the source of the data used in the text.
Response:
Added reference to data source in text - see Section 2, line 487.
- The Discussion section should better integrate previous studies, ideally by explicitly stating how this research has advanced compared to prior work.
Response:
This section was completed by comparing the results with other related works, and based on this, the contribution of our contribution was evaluated.
- The paper's writing style requires improvement, as some expressions are overly colloquial.
Response:
The authors are aware of this fact and tried to correct this style if possible.
All modifications have been highlighted in the text by yellow.
